# Distinct Neutralising and Complement-Fixing Antibody Responses Can Be Induced to the Same Antigen in Haemodialysis Patients After Immunisation with Different Vaccine Platforms

**DOI:** 10.3390/vaccines13010007

**Published:** 2024-12-25

**Authors:** Nadezhda Wall, Rachel Lamerton, Fiona Ashford, Marisol Perez-Toledo, Aleksandra Jasiulewicz, Gemma D. Banham, Maddy L. Newby, Sian E. Faustini, Alex G. Richter, Haresh Selvaskandan, Roseanne E. Billany, Sherna F. Adenwalla, Ian R. Henderson, Max Crispin, Matthew Graham-Brown, Lorraine Harper, Adam F. Cunningham

**Affiliations:** 1Institute of Applied Health Research, College of Medical and Dental Sciences, University of Birmingham, Birmingham B15 2TT, UK; 2University Hospitals Birmingham NHS Foundation Trust, Birmingham B15 2TH, UK; 3Institute of Immunology and Immunotherapy, College of Medical and Dental Sciences, University of Birmingham, Birmingham B15 2TT, UK; 4University Hospitals Coventry and Warwickshire NHS Trust, Coventry CV2 2DX, UK; 5School of Biological Sciences, University of Southampton, Southampton SO17 1BJ, UK; 6Department of Cardiovascular Sciences, University of Leicester, Leicester LE1 7RH, UK; 7National Institute for Health Research (NIHR) Leicester Biomedical Research Centre, University Hospitals of Leicester NHS Trust, University of Leicester, Leicester LE1 7RH, UK; 8Institute for Molecular Bioscience, The University of Queensland, Brisbane 4067, Australia

**Keywords:** haemodialysis, antibody, immune responses, SARS-CoV-2, vaccine

## Abstract

**Background/Objectives**: Generalised immune dysfunction in chronic kidney disease, especially in patients requiring haemodialysis (HD), significantly enhances the risk of severe infections. Vaccine-induced immunity is typically reduced in HD populations. The SARS-CoV-2 pandemic provided an opportunity to examine the magnitude and functionality of antibody responses in HD patients to a previously unencountered antigen—Spike (S)-glycoprotein—after vaccination with different vaccine platforms (viral vector (VV); mRNA (mRV)). **Methods:** We compared the total and functional anti-S antibody responses (cross-variant neutralisation and complement binding) in 187 HD patients and 43 healthy controls 21–28 days after serial immunisation. **Results**: After 2 doses of the same vaccine, HD patients had anti-S antibody levels and a complement binding capacity comparable to controls. However, 2 doses of mRV induced greater polyfunctional antibody responses than VV (defined by the presence of both complement binding and cross-variant neutralisation activity). Interestingly, an mRV boost after 2 doses of VV significantly enhanced antibody functionality in HD patients without a prior history of SARS-CoV-2 infection. **Conclusions**: HD patients can generate near-normal, functional antigen-specific antibody responses following serial vaccination to a novel antigen. Encouragingly, exploiting immunological memory by using mRNA vaccines and boosting may improve the success of vaccination strategies in this vulnerable patient population.

## 1. Introduction

Patients with chronic kidney disease (CKD), particularly those requiring haemodialysis (HD), have a significantly greater risk of infection and poorer infection-related outcomes than the general population, independent of immunosuppression, chemotherapy, or HIV infection [1,2,3,4]. The nature and cause(s) of the secondary immunodeficiency state associated with CKD remain incompletely understood [5]. Although defects in innate and adaptive immunity have been reported [6,7], CKD patients can maintain near-normal antibody responses to antigens that are likely first encountered years previously [8]. This suggests a state of dysfunctional immunity rather than abject immune failure.

Vaccination is a key intervention that is available to modify the risk of infection and associated morbidity/mortality across populations, particularly for those with CKD/HD. It also provides a model to understand immune function in patients. Multiple studies show impaired seroconversion following various conventional vaccines in CKD and HD populations, independent of immunosuppressive treatments [9,10], but this is not universal [11,12]. Surrogate measures for the success of vaccination within an individual can be quantitative measures, e.g., antibody levels, or qualitative measures such as antibody functionality, e.g., the capacity of an antibody to neutralise a pathogen and to fix and activate complement. Combining these read-outs can provide a more comprehensive assessment of immunity.

The SARS-CoV-2 pandemic had a devastating impact on patients with CKD, particularly those requiring HD, with 25–30% case fatality rates observed in pre-vaccination waves [13,14]. Nevertheless, surprisingly, some HD patients had encountered this novel pathogen and were able to generate SARS-CoV-2-specific immune responses without experiencing symptoms [15]. This meant that at least some HD patients maintained the capacity to induce de novo protective immunity to SARS-CoV-2. Critically, clinical outcomes and infection susceptibility in HD patients were significantly improved with the roll-out of SARS-CoV-2 vaccines [16,17,18,19,20,21], despite vaccine efficacy potentially being lower than in the general population [22,23]. Thus, immunity in CKD/HD patients can be positively modulated to improve infection-related outcomes. Studying antibody responses to the novel pathogen SARS-CoV-2 and its vaccines, all delivering the spike glycoprotein (S) via viral vector-based or mRNA technologies [24], provides an opportunity to examine how vaccine platforms and prior pathogen exposure influence aspects of antibody-associated immunity in these patients. In this study, we systematically characterise humoral immune responses in patients requiring HD by examining quantitative and qualitative antibody responses to the SARS-CoV-2 S antigen after serial vaccination with two different vaccine platforms, in the context of previous SARS-CoV-2 infection history.

## 2. Materials and Methods

### 2.1. Patient Selection and Data Collection

Patients established on HD were recruited from two UK centres—University Hospital Birmingham Foundation Trust (UHBFT) and University Hospitals of Leicester NHS Trust (UHL). Patients from 12 UHBFT satellite dialysis units were recruited to a prospective observational study of immune responses to SARS-CoV-2 vaccination (Coronavirus Immunological Analysis (CIA) study; ethical approval granted by North West-Preston Research Committee, ref 20/NW/0240). Control subjects were recruited from UHBFT and University of Birmingham employees (through internal advertising) and the general public as part of the CIA study. UHL patients requiring dialysis were recruited to the “Phenotyping Seroconversion Following Vaccination Against COVID-19 In Patients On Haemodialysis” study (ethical approval granted by West Midlands-Solihull Research Ethics Committee: ref 21/WM/0031). Only individuals over the age of 18 and eligible for SARS-CoV-2 vaccination were approached for participation in the study. Prisoners and individuals that did not have capacity as defined by the Mental Capacity Act were excluded. Written informed consent was obtained from all subjects involved in the study.

SARS-CoV-2 vaccination was performed in line with contemporaneous UK clinical guidelines [25]. National vaccination guidance was revised during the study period and recommended vaccine intervals were shortened. In most patients, the second vaccine doses were administered around 3 months after the first dose, while the third vaccine doses were administered around 6 months after the second dose.

UHBFT HD patients underwent weekly screening for SARS-CoV-2 infection via the PCR testing of nasopharyngeal swabs as part of standard clinical care. UHBFT HD patients that returned a positive PCR result were routinely dialysed in a central “COVID cohort” dialysis centre for 14 days, with regular blood tests and clinical reviews as a standard of care. UHL HD patients were only tested if they developed symptoms compatible with SARS-CoV-2 infection. No prospective data collection for the incidence of SARS-CoV-2 infection was performed in controls.

Demographics, SARS-CoV-2 vaccination status, and laboratory and clinical data were collected from electronic patient records. Immunosuppression was defined as the current use of immunosuppressant medication (e.g., prednisolone >5 mg per day or an equivalent dose of another steroid, tacrolimus, mycophenolate, or azathioprine), cyclophosphamide/methotrexate/plasma exchange in the last 6 months, or immunosuppressive monoclonal antibody in the last 12 months. Previous SARS-CoV-2 exposure was defined as a confirmed positive nasopharyngeal SARS-CoV-2 PCR prior to the start of the study and/or the detection of an anti-nucleocapsid (anti-N) antibody at study entry.

Some data from the UHL HD patient cohort have been published previously [26,27,28]. In this study, we have combined the previously reported neutralisation antibody data against the Wuhan, Delta, and Omicron variants from the UHL patients (n = 84) with previously unpublished data from a larger HD patient cohort from UHBFT (n = 103). We also report previously unpublished data on anti-S antibody responses in UHL patients following their second and third vaccine doses.

### 2.2. Serological Analysis

Serum samples were collected 21–28 days after the second SARS-CoV-2 vaccination in all subjects, and samples were collected 21–28 days after the third vaccine dose only in HD patients. Sera were analysed for antibodies directed against the SARS-CoV-2 Spike protein receptor-binding domain (S-RBD) and nucleocapsid (N) using an established automated electrochemiluminescence assay (Elecsys Anti-SARS-CoV-2 S and N, Roche Diagnostics International Ltd., Rotkreuz, Switzerland) [29]. Seropositivity was defined as anti-S levels greater than 0.8 U/mL. We chose to use a single anti-S antibody assay in this study as we have previously shown that, after three vaccine doses, patients requiring HD can generate comparable levels of antibody directed against S protein from different SARS-CoV-2 variants, including Delta and Omicron [30].

The binding of complement components to anti-S antibodies was assessed using a solid-phase C1q-binding assay and a C4b/3b/5b complement deposition assay, as described previously [31]. Briefly, 96-well microtiter plates were coated with 0.1 μg/mL HexaPro Wuhan SARS-CoV-2 S protein isolated from transfected HEK293F cells [32]. After blocking, diluted heat inactivated test sera (56 degrees C, 30 min) were added to the plate and incubated for 1 h at room temperature (RT). After washing, a standardised complement source (pooled SARS-CoV-2-negative normal human serum) was added to each well for 1 h at 37 degrees. Plates were incubated with monoclonal antibodies directed against complement proteins C1q, C3b, C4b, and C5b, and the signal was amplified using HRP-conjugated secondary antibodies and/or the PerkinElmer ELAST amplification kit (Waltham, MA, USA), as per the manufacturer’s instructions. Plates were developed using TMB Core (Bio-Rad, Watford, UK) and the reaction was stopped with H_2_SO_4_. Optical density (OD) was read at 450 nm using a SpectraMax ABS Plus plate reader (Molecular Devices UK Ltd., Wokingham, UK). The pooled mean OD of negative control wells for each assay was set as the “detection threshold” (0.5 for C1q; 0.1 for C4b, C3b, and C5b).

The anti-viral neutralising antibody (nAb) activity against SARS-CoV-2 was analysed in only HD patient sera, which was collected 21–28 days after the second and third vaccines using high-throughput live-virus neutralisation assays, as previously described [33]. Briefly, the neutralisation of live virus by serial dilutions of sera was evaluated using a SARS-CoV-2 isolate with a spike identical to the variants of interest. In this study, we examined the reactivity against three SARS-CoV-2 strains—Wuhan, Delta (B.1.617.2), and Omicron (B.1.1.529) isolates—with the latter being evaluated only in samples after 3 vaccine doses. The quantifiable assay range is from dilutions of 1:40 to 1:2560. Some sera display neutralising activity, but with an IC50 (the dilution at which 50% of infection is prevented) below this range. We defined “detectable inhibition” as sera displaying inhibition at the 1:40 dilution or higher.

### 2.3. Statistical Analysis

The statistical analysis of data was performed using Prism v9 (GraphPad, La Jolla, CA, USA) and SPSS v26 (IBM, Armonk, NY, USA). Two-sided tests were used throughout, with *p* values of 0.05 or less considered to be significant. Categorical variables were compared using Chi^2^ or Fisher’s exact tests. For continuous variables, unpaired comparisons were made using the Mann–Whitney U test, and Wilcoxon’s signed-rank test was used for paired comparisons. When comparing multiple groups, the Kruskal–Wallis test was used with Dunn’s post hoc multiple comparisons testing. Correlation analysis was performed using Spearman’s rank test. Multivariable linear and logistic regression modelling were used to examine the predictors of antibody quantity/quality and subsequent SARS-CoV-2 infection incidence. Non-parametric continuous variables were log_10_ transformed prior to their inclusion in regression modelling.

## 3. Results

### 3.1. Demographics/Descriptors of Study Population

The demographics and clinical parameters of study participants are described in Table 1. Patients requiring HD were significantly older, more likely to be male, and of non-white ethnicity compared to the controls. As expected, HD patients had more comorbidities than control patients, with a significantly greater prevalence of diabetes mellitus (DM; 41% versus nil in controls) and immunosuppression (IS; 9% in HD versus nil in controls). Patients requiring HD were broadly similar across the two UK study sites. A total of 18 HD patients developed post-vaccination SARS-CoV-2 infection—representing around 1 in 10 at both sites (Table 1).

A larger proportion of HD patients than controls received 2 doses of the AZD1222 vaccine than the BNT162b2 (hereafter described as viral vector (VV) and mRNA vaccines (mRV), respectively), but this did not reach statistical significance, and the time interval between first and second vaccine doses was similar between controls and HD patients. A higher proportion of HD patients had previous exposure to SARS-CoV-2 than controls (defined as either previous positive nasopharyngeal SARS-CoV-2 PCR and/or the detection of anti-nucleocapsid (anti-N) antibody at study entry), but this was not statistically significant. As such, age, gender, ethnicity, HD status, vaccine type, and previous SARS-CoV-2 exposure were included as co-variables in subsequent multivariable analyses. All HD patients received mRV as their third vaccine dose.

### 3.2. Patients Requiring HD Generate Similar Quantitative Antibody Responses as Controls Following Two Vaccine Doses

HD patients and controls had similar anti-S antibody levels at 21–28 days after a second vaccine dose when stratified by vaccine type and prior SARS-CoV-2 infection (Figure 1A–C; Table 2). Using the assay’s positive threshold of 0.8 U/mL for anti-S levels, all controls and almost all HD patients (n = 183/187; 98%) were seropositive after two vaccine doses.

Within both groups, post-vaccination anti-S antibody levels were generally higher in individuals vaccinated with two doses of mRV than VV, particularly in SARS-CoV-2-naïve individuals (Figure 1A,B). Previous infection with SARS-CoV-2 was associated with higher anti-S levels for HD patients receiving two vaccine doses, independent of vaccine type (Figure 1C). For analyses performed in HD patients, mRV, previous SARS-CoV-2 infection, and non-immunosuppressed status were significant predictors of higher anti-S antibody levels in a multivariable linear regression model that also included age, gender, ethnicity, and diabetes (Appendix A). No clinical/demographic parameters were significant predictors of non-response to two doses of vaccine in HD patients.

In summary, we have shown that after two vaccine doses, HD patients can have similar antigen-specific antibody levels to novel antigens as controls matched for vaccine type and previous pathogen exposure.

### 3.3. Vaccine Platform and HD Influence Fixation and Deposition of Complement by Antigen-Specific Antibody

As quantitative antigen-specific antibody responses after two vaccine doses were largely similar between controls and HD patients, we then compared surrogates of antigen-specific antibody functionality between the groups. One such surrogate is the binding and activation of complement components by antigen-specific antibody—the fixation of C1q and the deposition of downstream products of the complement cascade (C4b, C3b, and C5b) [31]. In seropositive individuals, the detection of complement component binding in our solid-phase assay positively correlated with the quantity of circulating anti-S antibody (Figure 2A–D).

Spearman’s rank correlation coefficient and *p* value shown for all data points. Black symbols denote controls, orange symbols denote HD patients, and filled symbols denote previous SARS-CoV-2 infection—see legend. Threshold level of antibody detection (“seropositivity”) are shown as a vertical dashed line (0.8 U/mL), complement binding assay negative thresholds are shown as horizontal dashed lines (0.5 for C1q and 0.1 for C3b/4b/5b).

As such, both mRV and previous SARS-CoV-2 infection were significantly associated with a greater binding of complement components by anti-S antibody (Table 2, Figure 3A–D). Although the majority of seropositive individuals (controls and HD patients) had detectable C1q fixation, SARS-CoV-2 infection-naïve VV recipients generally showed a lower binding of C3b-C5b to anti-S antibody than their mRV vaccinated counterparts (Table 2, Figure 3B–D). Indeed, a significantly lower proportion of SARS-CoV-2 infection-naïve VV recipients had detectable binding of all four complement components tested, both in control and HD cohorts (Table 2, Fisher’s exact *p* < 0.001).

The assay’s negative threshold (0.1 for C3b-C5b, 0.5 for C1q) is shown as a dashed line. A Kruskal–Wallis test was used for comparisons between groups (*p* < 0.05 for all complement components), with post hoc Dunn’s multiple comparison *p* values shown. The purple colour in panels I and J denotes comparisons of particular interest (described in main text); ns denotes *p* > 0.05.

Overall, when stratified by vaccine type and previous SARS-CoV-2 exposure, patients requiring HD exhibited greater heterogeneity than controls in the levels of complement binding detected (Figure 3A–D). Interestingly, despite having similar anti-S levels, SARS-CoV-2-naïve mRV vaccinated HD patients showed a significantly lower deposition of C4b and C3b than controls matched for vaccine type and previous pathogen exposure (Figure 3B,C—comparisons highlighted in purple).

In summary, we have shown that complement component fixation and deposition by antigen-specific antibody is highest in individuals that received two doses of mRV or those that have had a previous infection, with HD patients showing a greater variability in responses than controls.

### 3.4. In HD Patients, the mRNA Vaccine Induces a Broader Functionality of Antigen-Specific Antibodies than the Viral Vector Vaccine

Sera from HD patients can exhibit reduced neutralisation activity against SARS-CoV-2 when compared to healthy controls [26]. Neutralisation activity against related, but genetically distinct, SARS-CoV-2 variants reflects the breadth of immune responses induced and is a desirable feature of the antibody response to SARS-CoV-2 vaccination [34,35]. As such, we then assessed the neutralising activity of sera (nAb) against the Delta variant as a proxy of fragment antigen binding (Fab) antibody segment diversity and functionality.

Most patients requiring HD for whom data were available were able to neutralise the Wuhan SARS-CoV-2 variant after two doses of any SARS-CoV-2 vaccine (94%; n = 126 of 134; Table 2). For the Delta variant, the mRV platform was associated with a greater prevalence of detectable nAb than the VV vaccine in SARS-CoV-2 infection-naïve HD patients (Table 2; Appendix A; Fisher’s exact 2-tailed *p* < 0.0001), but no differences according to vaccine type were observed in those who had experienced SARS-CoV-2 infection. Delta nAb-positive HD patients had significantly higher anti-S levels than Delta nAb-negative patients (Appendix A).

We then cross-compared Delta variant neutralisation and the binding of complement components as surrogate measures of Fab and fragment crystallisable (Fc) segment antibody function, respectively. Surprisingly, detectable Delta nAb and complement component binding were not mutually inclusive in HD patients. This was most pronounced in SARS-CoV-2-naïve VV vaccinees, where almost half of the individuals with detectable Delta neutralising antibody did not have detectable binding of all four complement components tested (Table 2). This proportion was significantly lower than that seen in SARS-CoV-2-naïve mRV and VV vaccinees with previous SARS-CoV-2 infection (Table 2; Fisher’s exact 2-tailed *p* < 0.001 and *p* = 0.02, respectively).

Interestingly, the presence of highly functional anti-S antibody (either detectable Delta nAb and/or binding to all four complement components tested) was significantly associated with a reduced likelihood of post-vaccination SARS-CoV-2 infection, independent of age, gender, ethnicity, the presence of DM/immunosuppression, vaccine type, and HD centre (included due to different infection screening practises) (Appendix A).

Overall, vaccination with the mRV platform in HD patients is associated with greater antibody functionality than with VV, and the presence of highly functional antigen-specific antibody is associated with protection against post-vaccination infection.

### 3.5. mRNA Vaccination After Two Doses of a Viral Vector Vaccine Significantly Improves Antigen-Specific Antibody Function in Patients Requiring HD

During the study period, the HD population (as a clinically vulnerable group) were offered a third vaccine dose (booster) at 6 months after the completion of the primary (two dose) course, but this was not routinely carried out for controls [25]. This enabled us to compare the impact of receiving heterologous (VV followed by mRV) or homologous (mRV only) vaccines on quantitative and functional measures of circulating anti-S antibody in patients requiring HD.

All HD patients had detectable anti-S antibody after three doses of SARS-CoV-2 (Figure 4). The third vaccine dose significantly increased anti-S antibody levels in the viral vector vaccine groups, irrespective of previous SARS-CoV-2 infection history (Figure 4, Table 3). In HD patients who had previously received mRV, only the group without previous SARS-CoV-2 infection demonstrated an increase in antibody levels after the third vaccine dose (Figure 4, Table 3).

A third vaccine dose significantly increased the proportion of Delta nAb-positive individuals in VV recipients (Table 3), such that after three vaccine doses, there were no longer any vaccine type-associated differences in the capacity of HD patient sera to neutralise the Delta SARS-CoV-2 variant (Fisher’s exact *p* > 0.99). The presence of neutralising activity against the Omicron variant was observed in the majority of Delta nAb-positive individuals after three vaccine doses (97 of 106 sera; 92%; Appendix A). A third vaccine dose also increased complement component binding in the HD patient cohort. This was most pronounced in VV vaccinees, regardless of previous SARS-CoV-2 exposure, where the binding of C1q, C3b, C4b, and C5b was significantly higher after three doses of vaccine than after two (Table 3).

A third vaccine dose had a striking impact on the antibody functionality of VV vaccinees, as measured by dual capacity to neutralise the Delta variant and bind complement components (Figure 5A), particularly in SARS-CoV-2-naïve HD patients (Figure 5B). Here, the third vaccine increased the proportion of Delta-neutralising, complement-fixing antibody from just over 50% to 95%, a level similar to that seen in those that received triple mRV (Table 3). A similar pattern was seen when neutralising antibody directed against Omicron after three vaccine doses was considered alongside complement binding capacity (Appendix A).

In summary, an mRNA vaccine given after two doses of a VV vaccine significantly improves the quality of antigen-specific antibody in HD patients to levels comparable to individuals receiving mRNA vaccines alone.

## 4. Discussion

In this study, we examined the quantity and features of the quality of antigen-specific antibody following serial vaccination in patients requiring HD. The use of different vaccine platforms and the potential influence of infection provides us with the opportunity to understand how immunity develops to the same antigen presented in different immunological contexts.

Patients requiring HD can generate anti-S antibody after SARS-CoV-2 infection, which is associated with protection [36]. Nevertheless, after primary vaccination, quantitative humoral anti-S responses are typically lower than in controls [37,38,39,40]. This may be because HD patients, who are often B cell lymphopenic, have a less-diverse naïve B cell repertoire [41], and thus fewer B cells can potentially be recruited into primary responses, as observed after B cell depletion therapies [42]. Nevertheless, the recollection of humoral responses is largely preserved in HD patients [18,43,44,45]. This suggests that germinal centres are maintained and produce memory B cells that can respond upon subsequent antigen challenge. As such, HD patients can have immunological systems that can ultimately induce near-normal antibody responses, but this requires immunological memory recall, combined with optimised methods of antigen delivery.

The quantity of antigen-specific antibody and qualitative features of that antibody are frequently used as correlates of vaccine-associated immunity [46,47,48,49,50]. Fc functions of antibodies are conserved for when the anti-S antibody cross-reacts with different virus variants [51,52]; the engagement of the classical complement pathway by antibodies may contribute to this or be a proxy for the development of antibody functionality [53]. How the antigen is encountered may influence the quantitative and qualitative features of the antibody response induced. SARS-CoV-2-naïve HD patients generate significantly lower nAb titres to the reference (Wuhan) virus and other variants than controls after two doses of VV, but not after mRV. Moreover, here, we find that previous SARS-CoV-2 infection promotes cross-variant neutralising antibody responses in HD patients (to both Delta and Omicron), which has previously been observed in healthy individuals [54,55]. In combination, these findings emphasise the fact that antibody responses in HD patients can be “normal” or “near-normal” when the antigen is encountered in certain immunological contexts, even after heterologous boosting mRV [56]. The importance of this is that it indicates that vaccines can be tailored, either in how they are built and/or delivered, to improve the immune responses they provoke in HD patients and, potentially, the clinical benefit they confer.

Despite our HD cohort being relatively large, ethnically diverse, and representative of the wider HD patient population in terms of comorbidity, our study has limitations. Our smaller control group is significantly younger and different in gender/ethnic mix. In order to remove potential confounding effects, we included these variables in multivariable predictive models comparing controls and HD patients. Due to the speed of vaccine rollout, we did not capture primary responses to vaccines, as was originally envisaged. As third vaccine doses were not routinely offered to healthy individuals at the time of study, we were unable to collect third vaccine data for controls. We did not perform virus neutralisation assays on the control group as others have shown largely comparable neutralising responses in patients receiving HD treatment to controls, using the same assay [26]. Our focus in this study was to examine whether vaccine platform affected the quality and quantity of antibody responses in a patient cohort with secondary immunodeficiency.

In summary, we have found that HD patients induce antigen-specific antibody responses that differ based on how the antigen is encountered. Our results suggest that patients requiring HD can mount effective recall immune responses, and that mRNA vaccine platforms may potentially enhance the functionality of antibody responses in this patient population. As such, greater study of immune responses in this vulnerable patient population, particularly in relation to responses to mRV, is warranted. One area of immediate relevance for study is in relation to pathogens of clinical interest, where current vaccination strategies yield poor or inconsistent; for example, seasonal influenza and hepatitis B [9].

## Figures and Tables

**Figure 1 vaccines-13-00007-f001:**
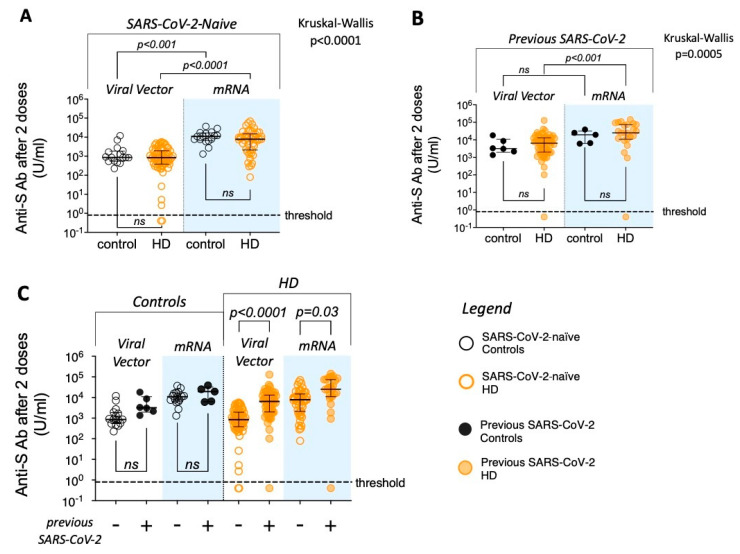
Antigen-specific antibody in sera collected 21–28 days after two doses of SARS-CoV-2 vaccine—comparison between controls and patients requiring HD. Comparisons of anti-S antibody levels between controls (black symbols) and patients requiring HD (orange symbols)—shown in legend. Filled symbols represent individuals with previous SARS-CoV-2 infection. Blue columns denote data from mRNA vaccinees. (**A**) Data from SARS-CoV-2-naïve individuals only; (**B**) data from individuals with previous SARS-CoV-2 infection only; (**C**) all data shown. Threshold level of antibody detection (“seropositivity”) shown as dashed line (0.8 U/mL). Kruskal–Wallis test used to compare groups in panels (**A**,**B**), with post hoc Dunn’s multiple comparison *p* values shown (ns denotes not significant; *p* > 0.05). The Mann–Whitney U test is used for pre-defined comparisons within groups in panel (**C**).

**Figure 2 vaccines-13-00007-f002:**
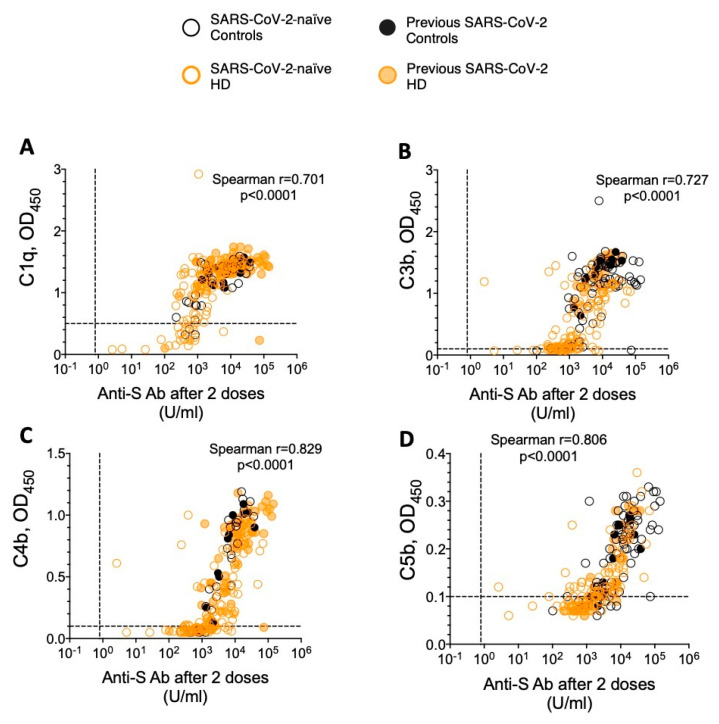
Correlation between anti-S antibody levels and the binding of complement components in controls and patients requiring HD. Data shown for all seropositive individuals—(**A**–**D**) correspond to C1q, C3b, C4b, and C5b, respectively.

**Figure 3 vaccines-13-00007-f003:**
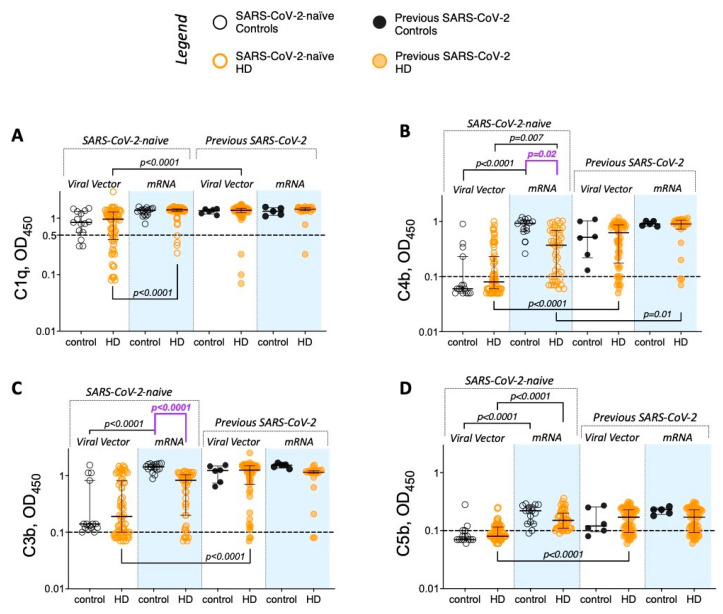
Complement component binding by antigen-specific antibody in sera collected 21–28 days after two doses of SARS-CoV-2 vaccine—comparison between controls and patients requiring HD. Comparisons shown between controls (black symbols) and patients requiring HD (orange symbols) with data grouped by vaccine type and previous SARS-CoV-2 exposure (filled symbols denote previous infection); (**A**–**D**) correspond to C1q, C3b, C4b, and C5b, respectively.

**Figure 4 vaccines-13-00007-f004:**
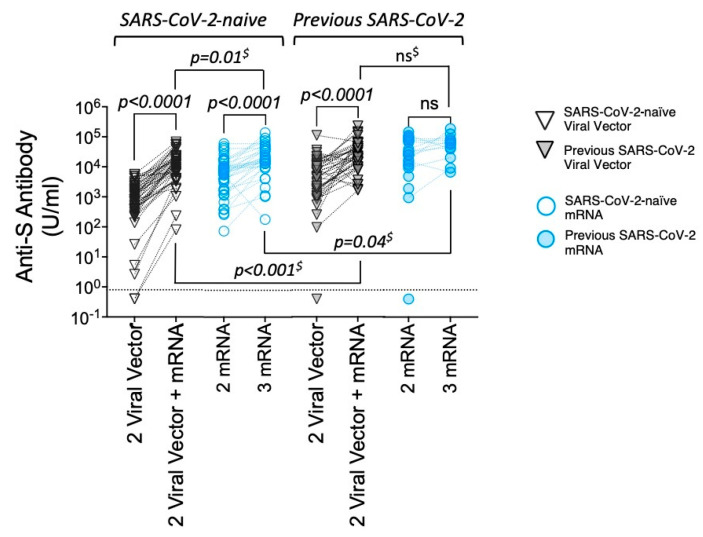
Anti-S antibody levels in sera of HD patients 21–28 days after two and three vaccine doses. Anti-S antibody levels in HD patients after two and three vaccine doses—comparisons by first vaccine type (grey—viral vector; blue—mRNA), split by previous SARS-CoV-2 infection (filled symbols denote previous infection)—see legend. Paired statistical comparisons performed between timepoints using Wilcoxon’s signed-rank test (post-dose two vs. post-dose three levels), and Mann–Whitney’s U test was used to compare antibody levels at the same timepoints by previous SARS-CoV-2 exposure (denoted by ^$^).

**Figure 5 vaccines-13-00007-f005:**
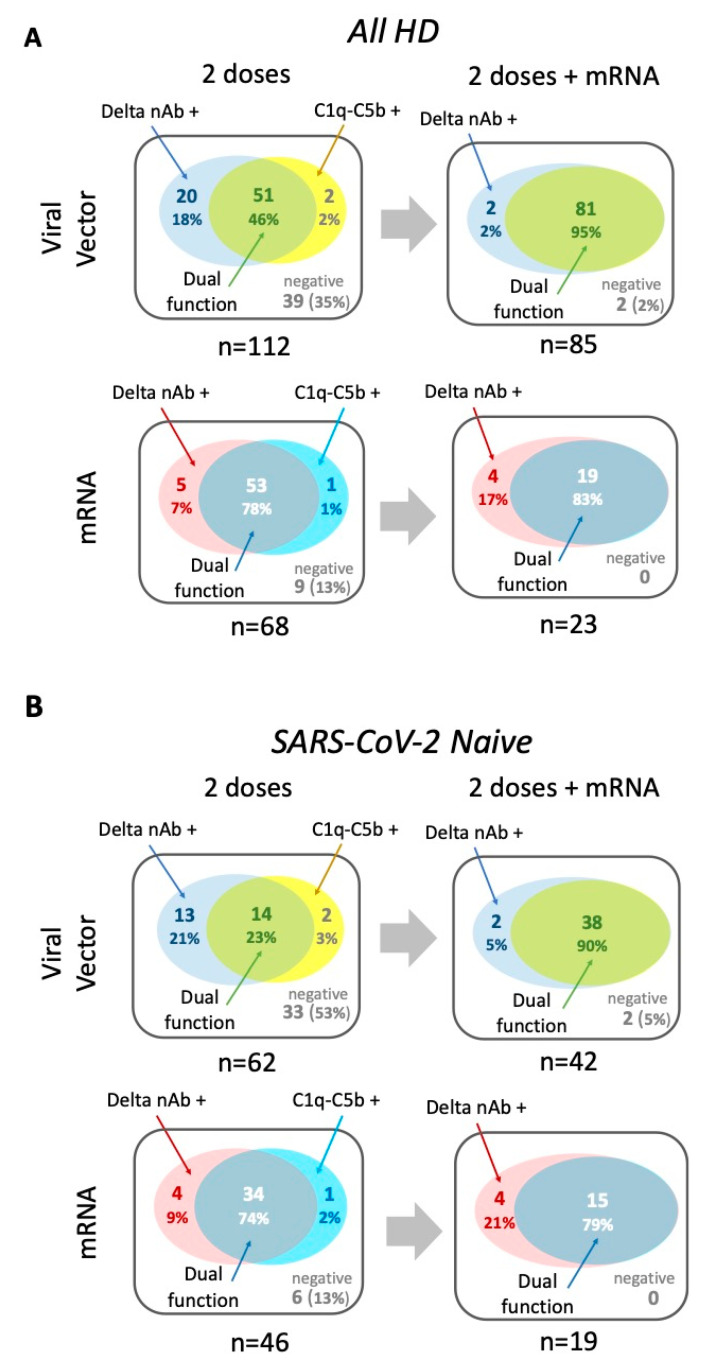
Antigen-specific antibody functionality in patients requiring HD—comparison between two and three vaccine doses. (**A**) Diagrammatic representation of antigen-specific antibody functionality after two and three vaccine doses for all HD patients—comparisons by vaccine type. Venn diagrams showing the overlap of neutralisation activity against Delta VoC (Delta nAb +) with the binding of all four complement components tested (C1q-C5b +) to denote antibody with dual function; n and % of total for whom data were available are shown. (**B**) As above, but for SARS-CoV-2-naïve HD patients only.

**Table 1 vaccines-13-00007-t001:** Demographics and clinical descriptors of the study population.

	Controls n = 43	HD n = 187	*p* Value
Age (years)	46 (17)	61 (15)	**<0.001**
Male gender, n (%)	18 (42%)	113 (60%)	**0.04** ^$^
Non-white ethnicity, n (%)	10 (23%)	99 (53%)	**<0.001** ^$^
Diabetes, n (%)	0 (0)	76 (41%)	**<0.001** ^$^
Immunosuppressed, n (%)	0 (0)	17 (9%)	**0.04** ^$^
Cause of renal disease, n (%)			
*Diabetic nephropathy*	-	57 (30%)	-
*Hypertensive nephropathy*	-	19 (10%)	-
*PKD/structural*	-	31 (17%)	-
*Immune mediated*	-	31 (17%)	-
*Other **	-	14 (7%)	-
*Unknown*	-	35 (19%)	-
Vaccine type **, n (%)			0.17 ^$^
*Viral vector*	22 (51%)	117 (63%)	-
*mRNA*	21 (49%)	70 (37%)	-
Time interval vaccine 1 to 2 (days)	80 (10)	82 (9)	0.08
Time interval vaccine 2 to 3 (days)	-	165 (32)	-
Previous SARS-CoV-2 infection, n (%)	11 (26%)	75 (40%)	0.08 ^$^
SARS-CoV-2 infection after vaccine 2, n (%)	-	18 (10%)	-

Data presented as median (IQR), unless stated otherwise. Statistical comparisons performed using a Mann–Whitney U test for continuous data and Fisher’s exact test (denoted by ^$^) for categorical data. *p* value <0.05 considered significant (highlighted in bold typeface). * Other causes of renal disease include multiple myeloma, trauma, and renal tuberculosis. ** denotes vaccine type given for first 2 doses. Abbreviations: HD—haemodialysis; PKD—polycystic kidney disease.

**Table 2 vaccines-13-00007-t002:** Serology after two vaccine doses—comparison between controls and HD patients.

	Viral Vector Controls	mRNA Controls		Viral Vector HD	mRNA HD	
All	n = 21	n = 21	*p* Value	n = 117	n = 70	*p* Value
Anti-S antibody level (U/mL)	1259 (2191)	11,111 (12,559)	**<0.001**	1788 (4580)	9921 (20,986)	**<0.001**
C1q fixation (OD)	1.12 (0.69)	1.36 (0.33)	**0.01**	1.24 (0.74)	1.41 (0.17)	**<0.001**
C3b deposition (OD)	0.14 (1.04)	1.45 (0.28)	**<0.001**	0.69 (1.20)	0.96 (0.84)	0.90
C4b deposition (OD)	0.07 (0.40)	0.91 (0.31)	**<0.001**	0.18 (0.65)	0.47 (0.77)	**<0.001**
C5b deposition (OD)	0.08 (0.04)	0.22 (0.10)	**<0.001**	0.10 (0.09)	0.17 (0.15)	**<0.001**
Detectable C1q-C5b binding *, % (n)	43% (9)	95% (20)	**<0.001**	46% (54)	80% (56)	**<0.001**
Wuhan nAb +ve *, % (n/total)	-	-	-	92% (95/103)	100% (31/31)	0.20
Delta nAb +ve *, % (n/total)	-	-	-	63% (71/112)	85% (58/68)	**0.002**
Delta nAb +ve with high complement binding *, % (n/total)	-	-	-	72% (51/71)	91% (53/58)	**<0.001**
**SARS-CoV-2-Naive**	**n = 15**	**n = 16**	***p*** **Value**	**n = 65**	**n = 47**	***p*** **Value**
Anti-S antibody level (U/mL)	860 (1323)	10,934 (10,171)	**<0.001**	841 (1550)	7818 (12,677)	**<0.001**
C1q fixation (OD)	0.84 (0.75)	1.37 (0.30)	**0.004**	0.96 (0.88)	1.40 (0.15)	**<0.001**
C3b deposition (OD)	0.14 (0.71)	1.43 (0.31)	**<0.001**	0.19 (0.71)	0.83 (0.84)	**0.03**
C4b deposition (OD)	0.06 (0.18)	0.93 (0.37)	**<0.001**	0.08 (0.17)	0.37 (0.59)	**<0.001**
C5b deposition (OD)	0.07 (0.03)	0.22 (0.13)	**<0.001**	0.08 (0.04)	0.15 (0.10)	**<0.001**
Detectable C1q-C5b binding *, % (n)	27% (4)	94% (15)	**<0.001**	25% (16)	76% (36)	**<0.001**
Wuhan nAb +ve *, % (n/total)	-	-	-	88% (51/58)	100% (23/23)	0.18
Delta nAb +ve *, % (n/total)	-	-	-	44% (27/62)	83% (38/46)	**<0.001**
Delta nAb +ve with high complement binding *, % (n/total)	-	-	-	52% (14/27)	89% (34/38)	**<0.001**
**Previous SARS-CoV-2 Infection**	**n = 6**	**n = 5**	***p*** **Value**	**n = 52**	**n = 23**	***p*** **Value**
Anti-S antibody level (U/mL)	3171 (8982)	19,338 (25,633)	0.06	6423 (11,131)	25,000 (63,169)	**<0.001**
C1q fixation (OD)	1.35 (0.26)	1.32 (0.41)	0.93	1.38 (0.25)	1.44 (0.17)	0.13
C3b deposition (OD)	1.23 (0.74)	1.53 (0.24)	0.06	1.25 (0.78)	1.15 (0.12)	0.18
C4b deposition (OD)	0.52 (0.80)	0.90 (0.22)	0.25	0.62 (0.69)	0.90 (0.34)	**0.006**
C5b deposition (OD)	0.12 (0.16)	0.23 (0.06)	0.33	0.17 (0.13)	0.26 (0.10)	**<0.001**
Detectable C1q-C5b binding *, % (n)	83% (5)	100% (5)	1.00	73% (38)	87% (20)	0.24
Wuhan nAb +ve *, % (n/total)	-	-	-	98% (44/45)	100% (8/8)	1.00
Delta nAb +ve *, % (n/total)	-	-	-	88% (44/50)	91% (20/22)	1.00
Delta nAb +ve with high complement binding *, % (n/total)	-	-	-	84% (37/44)	95% (19/20)	0.44

Initially, comparisons between the viral vector and mRNA vaccines were performed for the control and HD patients separately. The analyses were then repeated for the subgroups of patients that were SARS-CoV-2 naïve at the time of study entry, and those that had a previous SARS-CoV-2 infection. Data are presented as median (IQR) unless otherwise stated. For proportions—denominator (total n) is given where there is missing data. Mann–Whitney U test *p* values shown for continuous data; Fisher’s exact test *p* values shown for categorical data (denoted by *). *p* values <0.05 considered as significant and highlighted in bold typeface. Abbreviations: nAb—neutralising antibody activity (positivity defined as IC50 40 or greater).

**Table 3 vaccines-13-00007-t003:** Comparison of serology parameters after two and three vaccine doses in patients requiring HD stratified by previous SARS-CoV-2 exposure.

	Viral Vector			mRNA		
All HD Patients	2 Doses n = 117	2 Doses + mRNA n = 92	*p* Value	2 Doses n = 70	2 Doses + mRNA n = 40	*p* Value
anti-S Ab level (IU)	1788 (4580)	18,227 (36,914)	**<0.001**	9921 (20,986)	26,967 (45,573)	**<0.001**
fold change in anti-S Ab level	ref	9.9 (22.8)		ref	3.3 (6.3)	
C1q fixation (OD)	1.23 (0.74)	1.41 (0.28)	**<0.001**	1.41 (0.17)	1.40 (0.30)	0.90
C3b deposition (OD)	0.69 (1.20)	1.29 (0.24)	**<0.001**	0.96 (0.84)	1.22 (0.54)	**<0.001**
C4b deposition (OD)	0.18 (0.65)	0.90 (0.42)	**<0.001**	0.47 (0.77)	0.74 (0.66)	**<0.001**
C5b deposition (OD)	0.10 (0.09)	0.31 (0.13)	**<0.001**	0.17 (0.15)	0.24 (0.15)	**<0.001**
Detectable C1q-C5b binding *, % (n)	46% (54)	95% (87)	**<0.0001**	80% (56)	85% (34)	0.61
Delta nAb +ve *, % (n/total)	63% (71/112)	98% (83/85)	**<0.0001**	85% (58/68)	100% (23/23)	0.06
Delta nAb +ve with high complement binding *, % (n/total Delta nAb +ve)	72% (51/71)	98% (81/83)	**<0.0001**	91% (53/58)	83% (19/23)	0.26
**SARS-CoV-2-Naive**	**2 Doses** **n = 65**	**2 Doses + mRNA** **n = 46**	***p*** **Value**	**2 Doses** **n = 47**	**2 Doses + mRNA** **n = 28**	***p*** **Value**
anti-S Ab level (IU)	841 (1550)	12,401 (17,966)	**<0.001**	7818 (12,677)	20,800 (35,949)	**<0.001**
fold change in anti-S Ab level	ref	14.6 (33.1)		ref	3.7 (6.4)	
C1q fixation (OD)	0.96 (0.88)	1.37 (0.34)	**<0.001**	1.40 (0.15)	1.32 (0.34)	0.95
C3b deposition (OD)	0.19 (0.71)	1.16 (0.34)	**<0.001**	0.83 (0.84)	1.09 (0.76)	**<0.001**
C4b deposition (OD)	0.08 (0.17)	0.67 (0.55)	**<0.001**	0.37 (0.59)	0.46 (0.71)	**0.002**
C5b deposition (OD)	0.08 (0.04)	0.26 (0.17)	**<0.001**	0.15 (0.10)	0.17 (0.14)	**0.002**
Detectable C1q-C5b binding *, % (n)	25% (16)	89% (41)	**<0.0001**	77% (36)	82% (23)	0.77
Delta nAb +ve *, % (n/total)	44% (27/62)	95% (40/42)	**<0.0001**	83% (38/46)	100% (19/19)	0.09
Delta nAb +ve with high complement binding*, % (n/total Delta nAb +ve)	52% (14/27)	95% (38/40)	**<0.0001**	89% (34/38)	79% (15/19)	0.42
**Previous SARS-CoV-2 Infection**	**2 Doses** **n = 52**	**2 Doses + mRNA** **n = 46**	***p*** **Value**	**2 Doses** **n = 23**	**2 Doses + mRNA** **n = 12**	***p*** **Value**
anti-S Ab level (IU)	6423 (11,131)	30,650 (46,987)	**<0.001**	25,000 (63,169)	56,589 (57,269)	0.11
fold change in anti-S Ab level	ref	6.8 (9.9)		ref	2.2 (4.0)	
C1q fixation (OD)	1.38 (0.25)	1.45 (0.60)	**<0.001**	1.44 (0.17)	1.48 (0.26)	0.70
C3b deposition (OD)	1.25 (0.78)	1.33 (0.16)	**0.009**	1.15 (0.12)	1.36 (0.20)	**0.03**
C4b deposition (OD)	0.62 (0.69)	0.99 (0.21)	**<0.001**	0.90 (0.34)	0.93 (0.24)	0.21
C5b deposition (OD)	0.17 (0.13)	0.34 (0.06)	**<0.001**	0.26 (0.10)	0.28 (0.07)	0.10
Detectable C1q-C5b binding *, % (n)	73% (38)	100% (46)	**<0.0001**	87% (20)	92% (11)	0.99
Delta nAb +ve *, % (n/total)	88% (44/50)	100% (42/42)	**0.03**	91% (20/22)	100% (4/4)	0.99
Delta nAb +ve with high complement binding+, % (n/total Delta nAb +ve)	84% (37/44)	100% (42/42)	**0.01**	95% (19/20)	100% (4/4)	0.11

Initially, comparisons between two and three vaccine doses were performed separately for HD patients initially vaccinated with viral vector and mRNA vaccines. The analyses were then repeated for the subgroups of patients that were SARS-CoV-2 naïve at the time of the first vaccine, and those that had a previous SARS-CoV-2 infection. Data presented as median (IQR) unless otherwise stated. For proportions—denominator (total n) given where there is missing data. Mann–Whitney U test *p* values shown for continuous data, Fisher’s exact test *p* values shown for categorical data (denoted by *). *p* values <0.05 considered as significant and highlighted in bold typeface. Abbreviations: nAb—neutralising antibody activity (positivity defined as IC50 40 or greater).

## Data Availability

The original contributions presented in this study are included in the article/Appendix A. Further inquiries can be directed to the corresponding author.

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
