# Peer review of "Distinct Neutralising and Complement-Fixing Antibody Responses Can Be Induced to the Same Antigen in Haemodialysis Patients After Immunisation with Different Vaccine Platforms"

_vaccines, 2024, doi:10.3390/vaccines13010007_

Round 1

Reviewer 1 Report

Comments and Suggestions for Authors

Dear Authors 

The current manuscript aims to evaluate the distinct neutralizing and complement-fixing antibody responses among hemodialysis patients after immunization with different anti-

 SARS-CoV-2 vaccine platforms. Authors compared the total and functional anti-S antibody responses (cross-variant neutralization and complement binding) in 187 HD patients and 43 healthy controls after serial immunization. They noticed that HD patients can generate near-normal, functional antigen-specific antibody responses following serial viral vector and mRNA vaccines. This may improve the success of vaccination strategies in this vulnerable patient population.

This point is very interesting and well presented. I have just one comment: the vaccination schedule should be mentioned

  Best Regards

Author Response

Comments: The current manuscript aims to evaluate the distinct neutralizing and complement-fixing antibody responses among hemodialysis patients after immunization with different anti SARS-CoV-2 vaccine platforms. Authors compared the total and functional anti-S antibody responses (cross-variant neutralization and complement binding) in 187 HD patients and 43 healthy controls after serial immunization. They noticed that HD patients can generate near-normal, functional antigen-specific antibody responses following serial viral vector and mRNA vaccines. This may improve the success of vaccination strategies in this vulnerable patient population. This point is very interesting and well presented. I have just one comment: the vaccination schedule should be mentioned

Response: 

We thank the reviewer for their comments. We have now clarified the vaccination schedule in the Methods section (page 2, lines 92- 96). The median time intervals between vaccine doses are already included in Table 1 (80 and 82 days between 1st and 2nd doses in controls and HD patients, respectively, and 165 days between 2nd and 3rd doses in HD patients).

Reviewer 2 Report

Comments and Suggestions for Authors

This study exhibits significant methodological weaknesses that substantially limit its validity and generalizability. The control group has fundamental issues including demographic mismatches, inadequate size (43 vs 187 HD patients), and recruitment bias from healthcare workers. The temporal aspects are problematic with inconsistent sampling timepoints and lack of long-term follow-up. Methodologically, the study suffers from absence of neutralization assays in controls, inconsistent infection screening between centers, and non-standardized PCR testing protocols. Population studied is narrowly focused on UK centers, limiting global applicability, and shows inadequate representation of immunosuppressed patients. Additionally, the variant analysis is restricted primarily to Delta and Omicron, missing crucial data on emerging variants. These combined limitations severely impact the study's reliability and clinical applicability.

Author Response

Comments: 

This study exhibits significant methodological weaknesses that substantially limit its validity and generalizability. The control group has fundamental issues including demographic mismatches, inadequate size (43 vs 187 HD patients), and recruitment bias from healthcare workers. The temporal aspects are problematic with inconsistent sampling timepoints and lack of long-term follow-up. Methodologically, the study suffers from absence of neutralization assays in controls, inconsistent infection screening between centers, and non-standardized PCR testing protocols. Population studied is narrowly focused on UK centers, limiting global applicability, and shows inadequate representation of immunosuppressed patients. Additionally, the variant analysis is restricted primarily to Delta and Omicron, missing crucial data on emerging variants. These combined limitations severely impact the study's reliability and clinical applicability.

Response: 

We thank the reviewer for their comments. The study was performed under the constraints of the SARS-CoV-2 pandemic and with evolving national clinical guidance on groups eligible for vaccination and intervals between doses [The Green Book, Chapter 14a: COVID-19 - SARS-CoV-2, Public Health England, 2020]. We respectfully disagree that the group sizes are insufficient to identify differences. Our focus was not on the efficacy of the vaccines or long-term follow up, which have previously been described [doi: 10.1016/j.lanepe.2022.100478], but on the nature of the response to vaccines induced by HD patients. The overall IgG levels were similar between the groups, but the analysis on the complement-fixing capacity between the HC and HD sera (Figure 3 in manuscript) did show that they could be substantially different. Moreover, sera from the HD population did show different activities that associated with the vaccine platform used. This is consistent with a scenario whereby there can be minimal differences between the total IgG responses inducible in controls and HD patients (especially after boosting), but more notable differences in effector functions of those antibodies are seen that associate with the vaccine platform used to deliver the antigen. In other contexts, such critical observations have been key outcomes from higher throughput approaches such as system serology analyses [doi: 10.1111/imr.12503]. However, this is less well explored for sera from HD patients. Human studies with semi-synchronised delivery of a novel antigen through multiple licensed vaccine platforms are rare. This is particularly unusual in populations where timing of previous infections can be identified. Indeed, few licensed human vaccines even exist in multiple platforms to perform such studies, which makes the current study of greater interest to the vaccinology community. The reviewer also queries the technical approaches used by laboratories for PCR testing to detect infections. In the study centres involved, infection screening and PCR testing protocols were as per standard of care at the time and PCR testing was performed by accredited NHS laboratories. Our analyses were limited to Delta and Omicron as these were the main circulating variants at the time of study and were felt to be most clinically relevant. As they are genotypically different to the vaccine Wuhan variant, we feel they are adequate proxies of antibody diversity against other variants elicited by the vaccines. Immunosuppressed patients make up a small proportion of in-centre haemodialysis cohorts across the globe [doi: 10.1016/S2214-109X(23)00570-3]. It is well recognised that haemodialysis and chronic kidney disease are associated with a secondary immunodeficiency phenotype [doi: 10.1038/nrneph.2013.44]. As such, our study population is representative of the wider HD patient population and yields important insights about HD-associated immune dysfunction.

Reviewer 3 Report

Comments and Suggestions for Authors

It is a good paper. No improvements are required.

Author Response

Comments: It is a good paper. No improvements are required.

Response: We thank the reviewer for their kind comments.

Reviewer 4 Report

Comments and Suggestions for Authors

The authors have undertaken a difficult task - assessing the effectiveness of COVID-19 vaccination in dialysis patients. The manuscript is interesting, the study is well-designed, and the results are clearly presented. I have only one comment, and I did not find the answer in the text. In 24% of patients, the cause of chronic kidney disease is unknown/other. This is almost 1/4 of patients. I think an attempt should be made to explain these causes. For example, were there no poisonings (methanol, glycol, etc.) or dialysis resulting from drug overdose. In other words, was this a group of patients with chronic kidney disease, and was dialysis due to disease progression? Or was it an additional factor worsening the course?

Author Response

Comments: The authors have undertaken a difficult task - assessing the effectiveness of COVID-19 vaccination in dialysis patients. The manuscript is interesting, the study is well-designed, and the results are clearly presented. I have only one comment, and I did not find the answer in the text. In 24% of patients, the cause of chronic kidney disease is unknown/other. This is almost 1/4 of patients. I think an attempt should be made to explain these causes. For example, were there no poisonings (methanol, glycol, etc.) or dialysis resulting from drug overdose. In other words, was this a group of patients with chronic kidney disease, and was dialysis due to disease progression? Or was it an additional factor worsening the course?

Response: 

We thank the reviewer for their comments and for giving us the opportunity to add detail on this point as it is pertinent for contextualising the results. As the reviewer is aware chronic kidney disease is often an insidious progressive process with many patients presenting  at a relatively late stage of disease, when there is already a significant loss of renal function. Renal biopsies are often not performed, or if they are performed, then show only non-specific fibrotic changes that cannot be attributed to a single aetiology. In this report we have used the categories of primary renal disease as used in the UK Renal Registry Report. The proportion of patients with unknown/other cause of end-stage kidney disease in the assessed cohort here (24%) is similar to that reported across the UK (33% - 26th Annual Renal Registry Report, data up to 31/12/2022, available at www.ukkidney.org).  

Round 2

Reviewer 2 Report

Comments and Suggestions for Authors

All commentd have been addressed